# Modelling the Relationship between Rainfall and Mental Health Using Different Spatial and Temporal Units

**DOI:** 10.3390/ijerph18031312

**Published:** 2021-02-01

**Authors:** Matthew Yap, Matthew Tuson, Berwin Turlach, Bryan Boruff, David Whyatt

**Affiliations:** 1Medical School, University of Western Australia, Crawley 6009, Australia; matthew.yap@uwa.edu.au (M.Y.); matthew.tuson@uwa.edu.au (M.T.); 2Department of Mathematics and Statistics, University of Western Australia, Crawley 6009, Australia; berwin.turlach@uwa.edu.au; 3Department of Geography, University of Western Australia, Crawley 6009, Australia; bryan.boruff@uwa.edu.au; 4UWA School of Agriculture and Environment, University of Western Australia, Crawley 6009, Australia

**Keywords:** drought, rainfall, mental health, modifiable areal unit problem, modifiable temporal unit problem

## Abstract

Drought is thought to impact upon the mental health of agricultural communities, but studies of this relationship have reported inconsistent results. A source of inconsistency could be the aggregation of data by a single spatiotemporal unit of analysis, which induces the modifiable areal and temporal unit problems. To investigate this, mental health-related emergency department (MHED) presentations among residents of the Wheat Belt region of Western Australia, between 2002 and 2017, were examined. Average daily rainfall was used as a measure of drought. Associations between MHED presentations and rainfall were estimated based on various spatial aggregations of underlying data, at multiple temporal windows. Wide variation amongst results was observed. Despite this, two key features were found: Associations between MHED presentations and rainfall were generally positive when rainfall was measured in summer months (rate ratios up to 1.05 per 0.5 mm of daily rainfall) and generally negative when rainfall was measured in winter months (rate ratios as low as 0.96 per 0.5 mm of daily rainfall). These results demonstrate that the association between drought and mental health is quantifiable; however, the effect size is small and varies depending on the spatial and temporal arrangement of the underlying data. To improve understanding of this association, more studies should be undertaken with longer time spans and examining specific mental health outcomes, using a wide variety of spatiotemporal units.

## 1. Introduction

Mental health is a substantial burden on health systems around the world [1]. In Australia, mental ill-health represents 23% of the nonfatal burden of disease [2], and several recent studies have reported that rates of mental health-related emergency department (MHED) presentations are increasing [3,4].

One factor thought to contribute to mental illness is drought, particularly in agricultural communities [5,6]. Drought is expected to be an increasing hazard for both the agricultural industry and human health due to climate change, with Australia being one of the countries most susceptible to climate change-induced drought [5]. The causal relationship between drought and mental illness is characterized by a number of inter-related pathways, with the impact on agricultural productivity being a central component [6].

The economic impacts of drought in rural communities are well-established, as is the resulting loss of social networks and isolation that comes with stress, shame and other residents moving to the city. However, only a small number of published articles have quantified the relationship between drought and mental illness [6], and their results vary widely, despite all being from Australia. Further, most of the studies that do exist rely on survey data. Results from survey studies include: Higher odds of self-reported mental health problems in those living in drought-affected postcodes, compared to those living outside such areas [7]; and a higher score on the Kessler psychological distress scale among those in an area with a long and constant dryness pattern, compared to those in an area with a zero-to-moderate dryness pattern [8]. Studies using survey data have also found no relationship between drought and psychological distress in children [9] and women aged 53–58 [10]. Finally, contrasting results are sometimes reported in the same study, for example where psychological distress was found to be higher in areas with rainfall both above and below historical averages [11].

Survey data are known to be associated with recall and social desirability biases [12]. As an alternative, administrative data such as ED presentations have several qualities which make them useful for quantifying the relationship between drought and mental health. These include: Many years of data with specific timestamps that allow for various temporal aggregations; small-area residential information; and events that are countable and have clear and quantifiable economic consequences. Despite these qualities, few studies have used administrative data to examine the relationship between drought and mental health. One example is an analysis of suicide in rural New South Wales in Australia, between 1970 and 2007 [13]. That study reported an increased risk of suicide in males aged 10–29 and 30–49, and reduced risk in females, associated with increases in the Hutchinson Drought Index (i.e. increasing drought conditions).

An important aspect of the observed variation in published results is that, within the various pathways of causation, mental health is likely to be impacted by rainfall over different time scales. For example, mild psychological distress or exacerbations of an existing mental health condition might occur during a period of low rainfall, while severe depression might manifest after years of prolonged drought. Further, results may differ according to how time is segmented at a given scale. For example, mental health or rainfall could potentially be measured by years from January to December or from July to June, or by any other segmentation of time. Finally, a period of recent rainfall may impact mental health differently to one occurring further back in time (i.e. at a longer lag). Collectively, these problems are described by the modifiable temporal unit problem (MTUP) [14].

Analogous to its temporal counterpart, the modifiable areal unit problem (MAUP) describes how results may differ when based on alternative sets of aggregate-level spatial units (hereafter: ‘Zonations’) [15,16]. Previous studies, including those already cited, have examined data aggregated by single zonations such as ‘region’ [13] or postcode [7]. The results of those studies are consequently dependent on the chosen zonations. While this dependency was first described in 1934 [17], it is frequently ignored by modern-day geographers and public health experts [18]. The MAUP can manifest whenever zonations of different size (i.e. scale) or configuration are used [16].

Given the potential impact of the MTUP and MAUP, studies are needed that examine the association between mental health and rainfall using data aggregated by multiple time periods, lengths of time and spatial zonations. Such an approach may increase understanding of how variations and inconsistencies in previous studies’ results have occurred.

As such, this study will examine the association between MHED presentations and rainfall in the Wheat Belt region of Western Australia (WA), using various combinations of spatial and temporal units, and varying the difference in time between the measurement of MHED presentations and rainfall. In doing so, for different temporal combinations, results are reported that can be considered to be relatively independent of the spatial units on which they are based.

## 2. Materials and Methods

### 2.1. Geographical Study Area

The precise definition of the study area was decided in consultation with the WA Department of Primary Industry and Regional Development to align with dryland cropping as indicated by land use data [19]. The Wheat Belt represents the wheat-producing region of WA. It produces 50% of Australia’s wheat [20].

Specifically, the Wheat Belt region was defined to comprise the following Australian Bureau of Statistics (ABS) Statistical Areas (SAs) [21]: SA level 4: ‘Wheat Belt’; SA level 3: ‘Esperance’; SA level 2: ‘Northampton–Mullewa–Greenough’; ‘Geraldton’; ‘Geraldton–North’; ‘Geraldton–East’; ‘Geraldton–South’; ‘Irwin’; ‘Morawa’.

### 2.2. Population Data

ABS resident population data were obtained using the ABS TableBuilder tool, stratified by gender and either: SA level 1 (SA1), for the 2011 and 2016 Australian Censuses; or collection district, for the 2006 Census [22]. These data were standardized to 2016 SA1 boundaries using geographical correspondence ratios obtained from the ABS [21].

### 2.3. Health Data

MHED data were extracted from the WA Data Collections, specifically the Emergency Department Data Collection [23].

MHED presentations were defined to be those with either: (i) An International Classification of Diseases, Tenth Revision, Australian Modification (ICD-10-AM) diagnosis code beginning with ‘F’ (Mental and behavioural disorders) [24], or (ii) a Major Diagnostic Category (primarily used in regional WA) of 19 (Mental diseases and disorders) or 20 (Alcohol/drug use and alcohol/drug-induced organic mental disorders). All presentations to WA EDs (excluding presentations to one private metropolitan ED, from which patient-level data were not available) occurring between 1 January 2002 and 30 June 2017 and meeting the above diagnostic definition were included, if the patient’s SA1 of residence fell within the study area.

### 2.4. Rainfall Data

Daily rainfall data between 1 August 1999 and 30 June 2017, expressed as average precipitation per day in millimetres (mm), were obtained from the Australian Bureau of Meteorology. These data were available by a grid of cell size approximately 5 km by 5 km. For details on their preparation, see [25].

### 2.5. Data Preparation

The gridded rainfall data were mapped to SA1 using area-weighted averages, to match the spatial support of the health data. Then, data were aggregated by temporal windows of varying size and with varying endpoints. Specifically, MHED presentations were aggregated by windows of length 12 months, ending in December, October, August, June, April and February. These endpoints were chosen as a sample of potential changes to the position of the window. Population data for each window were derived by linear interpolation between census years and linear extrapolation for populations in 2002 to 2005 and 2017.

MHED windows were paired with rainfall windows of length 3 or 6 months, lagged by 0, 6, 12, 18 and 24 months from the end of the MHED windows. Rainfall windows longer than 6 months were not considered, because the present interest was in the effect of seasonal rainfall, which would be obscured by the use of larger windows.

To illustrate, Figure 1 shows two sets of temporal data arrangements. The first set of arrangements (left) shows a MHED window ending in December, and rainfall windows at lags of 0, 6 or 12 months. The second set of arrangements (right) shows a MHED window ending in October, and rainfall windows at the same three lags.

To allow for analysis at larger scales of population while ensuring that results were not dependent on any particular spatial aggregation, a system of different spatial aggregations was devised. Specifically, for each window pair, 26 different spatial zonations were examined: SA1s and 25 aggregate-level zonations of SA1s created using the freely available software AZTool [26,27]. The latter zonations were created based on five target population sizes: 500; 1000; 2500; 5000; and 7500, using the 2016 Census SA1 population data. Five zonations were created for each target population size. Units within these zonations contained the sum of the population and MHED events, and the mean rainfall for each day, calculated across their comprised SA1s.

For each window pair, separate analyses were carried out for males and females in order to elucidate gender-specific effects. Thus, 3120 different preparations of the data were created in total (Figure 2).

Within each dataset, each spatial unit was represented 15 times according to the number of possible MHED windows. Only MHED windows with start and endpoints within the study period were included.

### 2.6. Statistical Models

To estimate the association between rainfall and MHED presentations, allowing for geographical variability, a Besag–York–Mollié (BYM) Bayesian model [28] was fitted to each of the 3120 data preparations. These models were fitted using the Integrated Nested Laplace Approximation (INLA) approach via the INLA package [29,30] within R version 3.6.1 [31]. Specifically, the model used is given by:$$y_{i}\;\sim \;Poisson\left(e_{i}\theta_{i}\right)$$
$$\log\left(\theta_{i}\right)=\alpha +\beta \times {rainfall}_{i}+u_{i}+v_{i}$$
where $y_{i}$ and $e_{i}$ are the observed and expected MHED presentation counts in the ith geographic unit, respectively; $\theta_{i}$ is the standardized incidence ratio; β is the coefficient of the rainfall covariate; α is the model intercept; $v_{i}$ is an unstructured random effect; and $u_{i}$ is a spatial random effect modelled with a conditional autoregressive distribution. Prior distributions for $u_{i}$ and $v_{i}$ were specified as:$$v_{i}\;\sim \;N\left(0,\sigma_{v}^{2}\right)$$
$$u_{i}|u_{-i}\;\sim \;N({\bar{u}}_{\delta_{i}}\;|\;\frac{\sigma_{u}^{2}}{n_{\delta_{i}}})$$
where
$${\bar{u}}_{\delta_{i}}=\;\frac{1}{n_{\delta_{i}}}\sum_{j\in \delta_{i}}u_{j}$$
and $\delta_{i}$ is the set of $n_{\delta_{i}}$ areas neighbouring area i. Following a sensitivity analysis undertaken by Cramb and colleagues [32], hyperprior distributions for $\sigma_{v}^{2}$ and $\sigma_{u}^{2}$ were specified as:$$\sigma_{v}^{2}\;\sim \;InvGamma\left(0.001,0.001\right)$$
$$\sigma_{u}^{2}\;\sim \;InvGamma\left(0.1,0.1\right)$$

By default, α and β are assigned independent prior normal distributions with zero mean and large variance [33].

### 2.7. Ethical Approval

The study was approved by the Western Australia Department of Health Human Research Ethics Committee under the Project Reference Number RGS0000002799 on 17 June 2019.

## 3. Results

### 3.1. Summary Characteristics

The characteristics of the study population are summarized in Table 1. Across calendar years, there was an annual mean population size of approximately 190,000 residents, who experienced an average of 3700 MHED presentations per year. On average, the socioeconomic status (SES) of SA1s within the study region was in the bottom third of that for all WA SA1s. The rate of MHED presentations increased from 15 to 23 per 1000 individuals between 2002 and 2017, when aggregated by calendar year (Figure 3). Note: As data from only the first half of 2017 were available, the rate for that year was calculated based on half the 2017 population size for illustration in Figure 3.

Figure 4 and Figure 5 show the mean annual and seasonal rainfall of the study area, between 1 January 1999 and 30 June 2017. The dashed line before the year 2002 indicates the date of earliest available data for MHED presentations. Average rainfall for the study area varied widely from 180 to 455 mm per calendar year, with a daily mean of 0.81 mm (SD 0.23). Mean daily summer rainfall (December–February) was 0.43 mm (SD 0.47), and mean daily winter rainfall (June–August) was 1.50 mm (SD 0.49).

### 3.2. Model Results

From each model, the coefficient for rainfall was exponentiated to derive a rate ratio (RR). A 95% credible interval for each RR was derived in the same manner. The standard deviation of average daily winter and summer rainfall (approximately 0.5 mm) was used to scale the RRs so that they are interpretable as representing the estimated posterior mean change in the RR of MHED presentations for each standard deviation increase in rainfall.

Figure 6 and Figure 7 show the RRs and credible intervals derived from each of the 3120 model fits. Figure 6 shows estimates for females, and Figure 7 shows estimates for males. In each figure, the estimates are arranged in panels, with each panel representing a temporal arrangement. The columns indicate the MHED window endpoints and the length of the rainfall period. The rows represent the lag between the MHED and rainfall windows. In each panel, different shades of blue are used to differentiate between the estimates of different zonations; the SA1 estimates are coloured black. Note that the x-axis values correspond to approximately half the target population sizes used to create the various zonations, as that creation was undertaken prior to the stratification by gender.

Figure 6 and Figure 7 show that, for a given temporal arrangement, different RRs are obtained using the different spatial zonations. In some arrangements, the RRs consistently indicate either a positive or negative association between MHED presentations and rainfall. For example, the top-right panel of Figure 7 (‘December 6, 0’; i.e. MHED window ending in December and a 6-month rainfall window with lag 0) indicates a consistently negative association, though the SA1 effect size is approximately double that of the higher-scale zonations. Similarly, the lower-left panel of Figure 7 (‘February 3, 24’; i.e. MHED window ending in February and a 3 month rainfall window with lag 24 months) indicates a consistently positive relationship. On the other hand, in some panels, inconsistent associations are observed; such cases are characterized by the 95% credible interval of at least one RR containing the value 1, or the direction of at least one RR contradicting that of another. For example, the top-right panel of Figure 6 shows a negative SA1-level association but positive estimates for the higher-scale zonations.

Figure 8 summarizes the results in Figure 6 and Figure 7. The arrangement of panels is the same as in those figures, but with a single coloured tile representing each panel. The tiles are coloured blue, green or red, according to whether their RRs were consistently negative, inconsistent or consistently positive, respectively. The strength of shading in each panel represents the mean effect size among the RRs it summarizes (larger absolute mean effect sizes are shaded darker). There are two features of interest in this summary. The first is a negative relationship between rainfall and MHED presentation rates, with 6 months of lag when the MHED window ends in February and April, and with 0 and 12 months of lag when the MHED window ends in August and October. These months of rainfall correspond to winter in Australia, which is the growing season. The relationship is observed for both males and females. Among the relevant estimates, RRs were as low as 0.96 for each 0.5 mm increase in daily rainfall. The second feature observed is a positive relationship between rainfall and MHED presentation rates, with 12 and 24 months of lag when the MHED window ends in February and April, and with 6 and 18 months of lag when the MHED window ends in October, though the latter effect is more consistent for the female population. These months correspond to the Australian summer, which is harvest season for grains. Among the relevant estimates, RRs were as high as 1.05 for each 0.5 mm increase in daily rainfall.

## 4. Discussion

Consistent with reports on climate change in WA, this study has shown that rainfall in the Wheat Belt region varies from year to year, especially during the summer [34]. Further, rates of MHED presentations within this region have been shown to be increasing over the past ten years, and were consistently higher than those previously reported for metropolitan WA over the same time period [3]. In this context, the present study has focused on estimating the association between rainfall and mental health using different spatial and temporal preparations of available data. In doing so, consistent with the literature of the MAUP and MTUP, highly variable estimates were observed; for example, for some temporal data preparations, estimates based on different spatial units were opposed with respect to their direction. Such results are independent of the type of statistical model used, and may explain the inconsistent findings of past studies, which have measured the association between rainfall and mental health using just one or two different preparations.

However, notwithstanding the observed variation in estimates, two key findings were noted: Associations between rates of MHED presentations and rainfall were generally positive when rainfall was measured in summer months, and generally negative when rainfall was measured in winter months. These relationships summarize estimates from models fitted to a wide range of spatial scales and zonations; as such, they can be considered robust in the sense that they are not detectably affected by the MAUP. However, their effect sizes were relatively small, with minimum and maximum RRs of 0.96 and 1.05 per standard deviation of daily rainfall, respectively.

The negative relationship between MHED presentation rates and rainfall measured in winter months (i.e. between May and October) is consistent with the rainfall requirements of the phenology of wheat, which is the dominant grain in the study area [35]. Reduced rainfall during winter months is likely to lead to reduced yields, and consequently increased financial stress among both farmers and the communities that depend on their economic input [6]. Beyond these direct impacts, prolonged drought over successive winters might result in a reduction in employment opportunities. Residents may move to the city or larger towns, leading to reductions in investment and social infrastructure. Further, the distress associated with observing the environmental degradation and suffering of livestock can have a large emotional toll [36]. Conversely, increased rainfall during winter months is likely to lead to beneficial impacts on grain yields, and may improve moisture availability for the following year, and consequently also mental health.

Additional rainfall during the harvest season (summer months) may also place financial strain on farmers, as grain quality can be degraded and heavy rainfall can cause erosion, loss of soil nutrients, damage to property and loss of livestock. Another possible explanation for the harmful impacts of summer rainfall is that such rain may necessitate the removal of weeds at a time when most farmers are usually on holiday; such overwork may ultimately result in MHED presentations to the local hospital. An influx of approximately 2000 seasonal workers occurs each year during harvest season [37], and may be thought to contribute to higher rates of MHED presentations; however, data used for the present study only included patients whose primary residential address was within the Wheat Belt region. Therefore, presentations by seasonal workers are unlikely to be a major driver of the observed relationship, unless their temporary rural residence was recorded as their address. It is also possible that there is an association between reduced winter rainfall and increased summer rainfall, so the observed relationship between increased rates of MHED presentations and summer rainfall may be confounded by decreased winter rainfall. More work is needed to identify the mechanism behind this finding.

As with all observational studies, interpretations of the associations reported here as representations of causal relationships must be made with caution. The present study is limited to observing associations between rainfall and MHED presentations using a simple time-stratified approach [38]. Attribution of increased rates of MHED presentations to changes in rainfall may be better approached using more complex concepts such as cointegration [39], as is often done in complex time series modelling; however, this would require a longer period of ED data than was available here. Excessive heat and frost may also reduce grain yields [40,41], which may in turn influence the mental health of the rural population. Apart from such climactic factors, population covariates such as age, socio-economic status and ethnicity may influence the mental health of farmers and the wider agricultural community. For example, the aging of the population may contribute to higher rates of ED presentations for diagnoses such as dementia. However, such factors are unlikely to impact the seasonal effects of rainfall observed in this study. Regardless, these factors were intentionally not considered in order to observe unadjusted relationships and maintain focus on the use of different spatial and temporal data preparations. This study has also used the concept of drought and rainfall interchangeably, though they are different phenomena. Drought has more than 150 definitions within the literature [42], and using a linear scale of rainfall in millimetres may be over-simplistic when applied to the impact upon the mental health of a population. Taking into account all these limitations, a cautious interpretation of the results reported here might consider variations in rainfall as representing changes in weather patterns.

As we have noted, the relationship between rainfall, or lack thereof, and mental health is complex, and follows a number of pathways. Further, these pathways differ by the scale of the affected population and the length of time before impacts are experienced [6]. Adding to this complexity is the definition of mental health. Within the present study, presentations related to the use of drugs and alcohol are grouped with presentations for anxiety, depression and psychosis. Given these issues, it is encouraging that any significant associations were observed at all. Future studies of the relationship between drought and mental health, or of similar relationships, must seek to disentangle these complications, and should incorporate examination of important intermediate factors such as grain yield and agricultural income. Future studies would also benefit from having sufficient temporal data on mental health outcomes to identify patterns that are consistent and follow plausible pathways.

It has previously been suggested that the MAUP can effectively be avoided by examining associations estimated based on data aggregated by a ‘minimal spatial unit’ [15]. However, due to the complexity of relationships by which rainfall may potentially impact upon mental health, it is unclear how such a unit might be defined in the present case. For example, mental health may be impacted by rainfall at the level of individual farmers, families, communities or regions with shared ports. Therefore, all spatial scales are potentially valid according to the theorized mechanisms that link mental health and rainfall.

As with mental health and drought, ecological studies of associations between a wide range of outcomes and geographic variables are commonly used to build evidence bases and direct policy. Examples of such studies include those examining the association between multiple diseases and Vitamin D exposure [43], or crime and socioeconomic status [44]. However, due to the MAUP, those studies are dependent upon the geographical unit used for analysis [14]. The methodology used within the present study is relevant to all such ecological studies.

Finally, the findings in this study have arisen from the context of the Western Australian Wheat Belt region. While the study area is broadly representative of the greater Australian agricultural community, drought-related policies in WA are not the same as for other states. For example, concessional loans for drought-stricken farmers in WA were ended in 2005 and the Exceptional Circumstances policy was ended in 2009. These assistance-based policies were considered to hinder self-reliance and adaptation, and were replaced by an increased focus on education and resilience measures for agricultural communities, including exit grants to assist farmers to leave the industry [45]. Further, farmers in other continents may not have the same access to climate and agricultural technology, and differences in social support systems are likely to modify the impact of changes in rainfall on their mental health.

## 5. Conclusions

The association between rainfall or drought and mental health is quantifiable; however, the effect size is small and varies depending on how the available data are spatially and temporally arranged. This observation may explain mixed results reported in existing literature on this topic. Nevertheless, in agreement with the literature on the health effects of drought, higher rates of MHED presentations were found to be associated with lower quantities of rainfall in winter months. By contrast, an unexpected finding is that higher rates of MHED presentations were associated with increased rainfall in summer months. To corroborate and understand these findings, more work should be undertaken over longer time spans, examining specific mental health outcomes. The use of multiple spatial and temporal units for analysis, as in the present study, will be essential in this endeavour.

## Figures and Tables

**Figure 1 ijerph-18-01312-f001:**
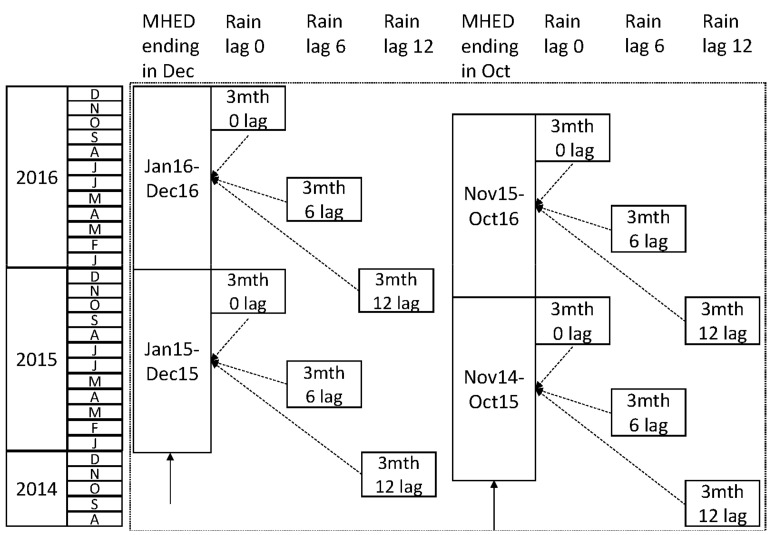
Example temporal arrangements. MHED: mental health-related emergency department; mth: month.

**Figure 2 ijerph-18-01312-f002:**
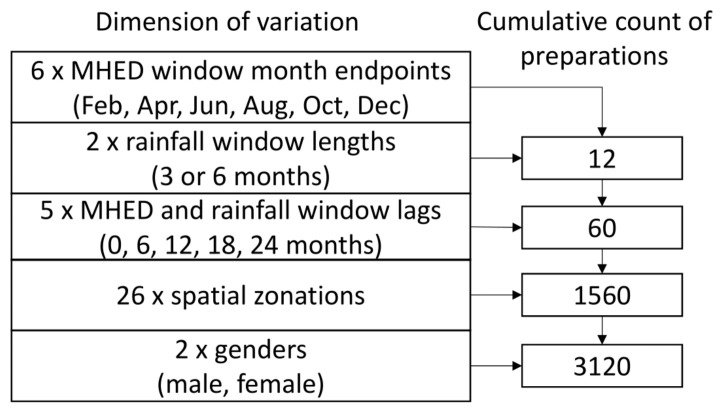
Summarized count of data preparations based on variations in time, space and gender. MHED: Mental health-related emergency department.

**Figure 3 ijerph-18-01312-f003:**
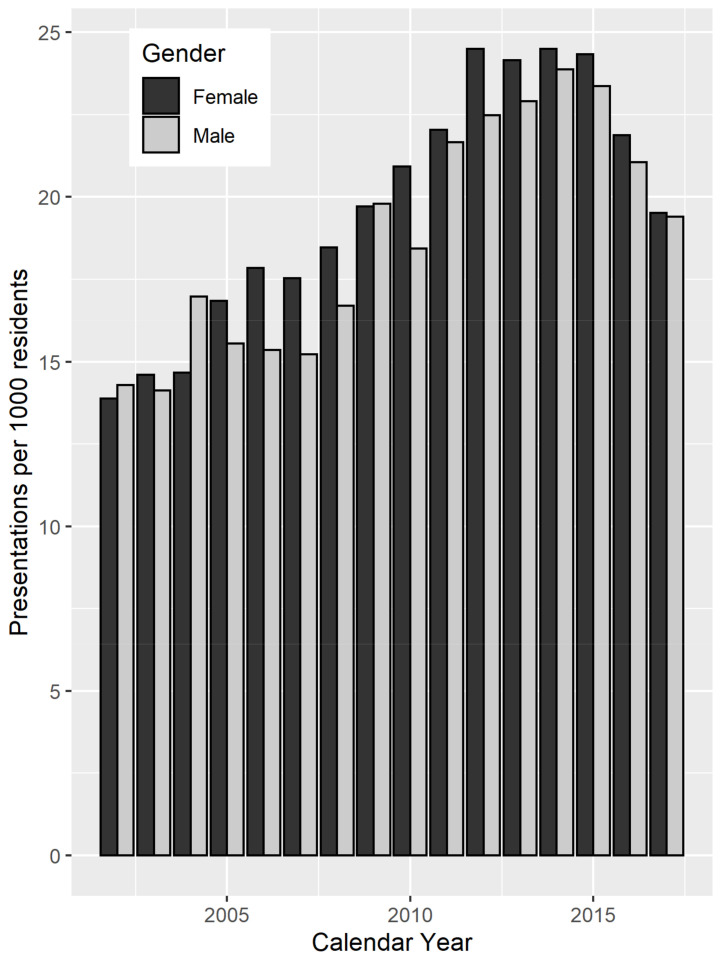
Annual rate of MHED presentations per 1000 residents, stratified by gender.

**Figure 4 ijerph-18-01312-f004:**
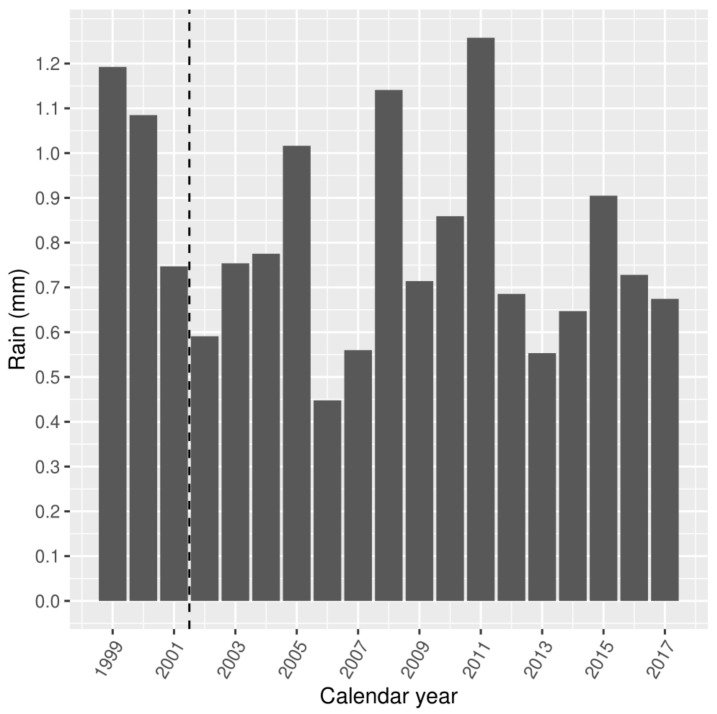
Mean daily rainfall (mm) by calendar year.

**Figure 5 ijerph-18-01312-f005:**
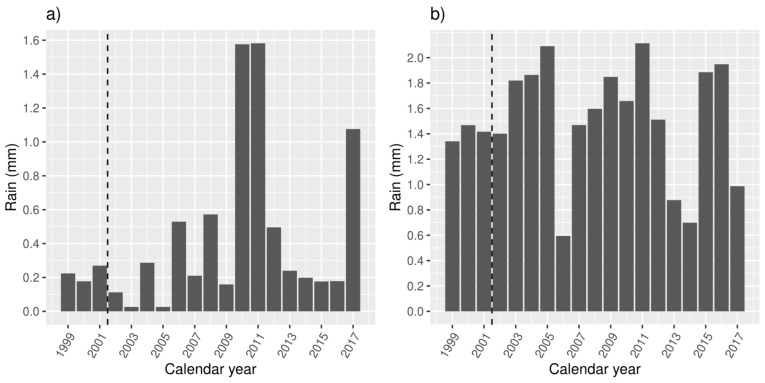
Mean daily summer and winter rainfall (mm) by year. (**a**) Summer (December–February); (**b**) winter (June–August). The dashed line indicates the beginning of available MHED data.

**Figure 6 ijerph-18-01312-f006:**
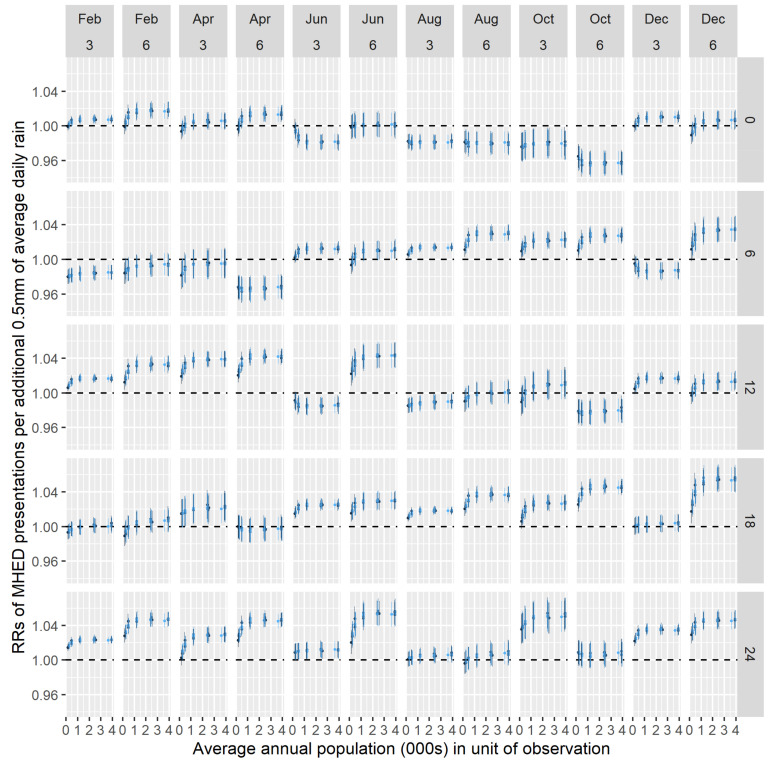
Rate ratios (RRs) (**dots**) and 95% credible intervals (**lines**) from models fitted to different spatial and temporal data preparations (females). Different shades of blue indicate estimates for different spatial zonations at each scale, in each panel. Black dots and lines indicate results from statistical area level 1 (SA1)-level analyses. MHED: mental health-related emergency department.

**Figure 7 ijerph-18-01312-f007:**
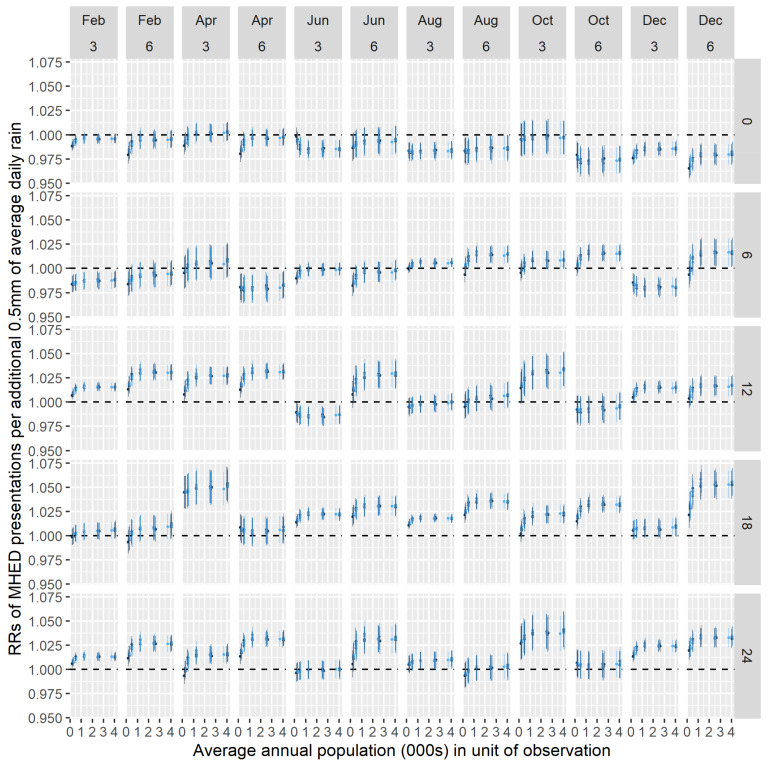
RRs (**dots**) and 95% credible intervals (**lines**) from models fitted to different spatial and temporal data preparations (males). Different shades of blue indicate estimates for different spatial zonations at each scale, in each panel. Black dots and lines indicate results from SA1-level analyses. MHED: mental health-related emergency department.

**Figure 8 ijerph-18-01312-f008:**
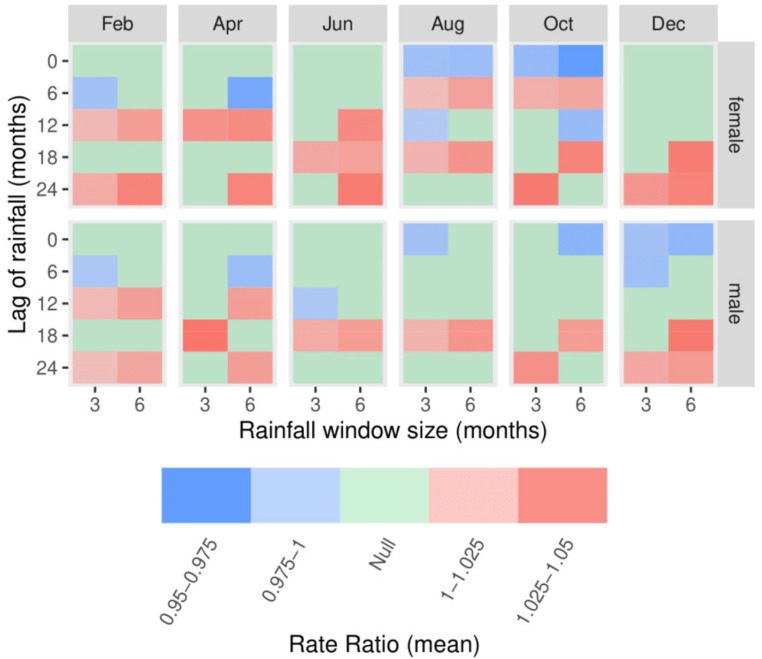
Mean RRs amongst regressions on all spatial units.

**Table 1 ijerph-18-01312-t001:** Characteristics of the Wheat Belt population from 1 January 2002 to 30 June 2017.

Resident Characteristics	
Mean age (SD)	38.5 (1.4)
Mean percentile SES (SD)	33.3 (4.6)
Mean population per square km (SD)	0.6 (0.02)
Percent male	50.8
Mean annual resident population (SD)	188,001 (5040)
**MHED Characteristics**	
Mean age of patients (SD)	36.9 (16.6)
Percent male	49.6
Mean annual MHED presentations (SD)	3682 (822)

SD: standard deviation; SES: socio-economic status; MHED: mental health-related emergency department.

## Data Availability

Restrictions apply to the availability of health data examined in this study. Health data were obtained from the Department of Health, Western Australia, and are available from the authors with the permission of the Department of Health, Western Australia.

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
