# Peer review of "Modelling the Relationship between Rainfall and Mental Health Using Different Spatial and Temporal Units"

_ijerph, 2021, doi:10.3390/ijerph18031312_

Round 1

Reviewer 1 Report

This research is interesting to investigate the relationship between rainfall (or drought) and mental health.

1 More information on the BYM models used in this research should be provided, including prior data for these Bayesian models. Moreover, the DIC (Deviance Information Criterion) from the Bayesian models should be discussed to assess model accuracy.

2 More cautions should be taken when explaining the Bayesian results since there may be hidden variables or factors, which are not considered in the research.

Reviewer 2 Report

This is a comprehensive analysis to eliminate the modified spatial and temporal unit problems underlying the inconsistent notions from previous studies investigating the causal relationships between rainfall and mental health issues in Western Australia Wheat belt. It is highly appreciated that the authors have identified the consistent relationships between decreased winter (growing season) rainfall and MHED presentations and additional increased summer (harvesting season) rainfall and MHED presentation with careful investigation of time lags and ending points of visits. This methodological approach enables to assess and compare the results using different target time and area unit.

I appreciate if the authors could confirm the limitation of this study, that the mean age of the target area is very young (38.5 y.o.) and the population density is very scarce (0.6/square km) which represents the style of Australian agriculture. Though the population was increasing in early 2000s, the population seems to be rapidly declining after 2014. The global agricultural population may face the aging and decreasing workforce population. Aging of the population may induce another impact in MHED visits.

It is also appreciated if the authors could provide any counter measures to cope with the rainfall induced mental health issues.

Although most of the necessary description is written in the manuscript, the figure legends could be a bit reader friendly as described below:

Fig. 1: To facilitate the understandability, Fig. 1 should have the time scale at vertical scale like, 2016, 2015, 2014 with months as smaller scale marks from up to down. I think it is confusing recent year comes first while the arrow goes down to show the temporal direction.

Fig. 5: Add explanation of dashed line.

Fig. 6 and 7: The title and the vertical axis may be better by adding "RRs of MHED presentation" to increase the understandability. Spelling out and the brief explanation of RR (ex. Exponential of the coefficient for rainfall (mm) to MHED presentation). Explain Black and Blue dots and bars.

Fig. 8: You have arranged Fig. 7 by male, female and Fig. 3 and 8 by female, male orders. I might suggest the consistency of expression with caution not compromising the manuscript. Add the color legend to Fig. 8.

Reviewer 3 Report

I read your article with interest and find your topic complex, but fascinating. Overall, if I understood you rightly, you were interested in the association between mental health presentations at Western Australian health centers and rainfall levels. Your overall results were inconclusive as indeed, one might expect them to be given the fact that, as you point out, 1) you really did not measure the effects of drought and 2) you included the effects of both drug use and anxiety, depression and psychosis in your date. Given the confounding array of variables that can contribute to each of these distinct categories, I was not surprised that you did not discover clear causal paths. This said, I came away with two larger questions that your piece as yet leaves unaddressed as well. First, when you argue that "drought is thought to impact upon mental health," you do not say who thinks so although the relationship(s) surely make some intuitive sense, though whether drought alone would cause depression or psychosis remains more than unclear. So, I wonder if it might make some sense to delimit clear ways in which drought could contribute to known triggers for these specific illnesses and thereby tie your effort more closely to something that could have clear import for public policy. Just now,it is unclear to me that you have learned anything that policymakers did not already know or could, in fact, employ to inform their work. Likewise, and again as you rightly point up, either higher or lower rainfall levels can have deleterious effects for farmers. What I know no more about after reading your piece is how that plays out in its effects on health outcomes in which you are interested, except that there is an association between them. These observations in turn lead me to wonder about two things. First, whether this line of inquiry represents something of a blind alley. That is,  even as you suggest more such aggregate scale research over longer periods. it is unclear to me what those efforts will show in these terms ? I wonder instead, as I have intimated, if it might make make more sense to adopt an alternate analytical approach that might allow you to tie rainfall and drought to specific mental health conditions in ways that you now cannot. Once those relationships are explored you could perhaps develop models of the sort that now intrigue on that data.

Round 2

Reviewer 3 Report

I suggest that you highlight the fact that your specific contribution and goal in this piece is methodological. You now reach that point on page 3, line 110. It would be helpful to ensure your readers understand it as your focus in your abstract and introduction and thereafter that you make it central and its character clear throughout.

Author Response

Reviewer comments:

I suggest that you highlight the fact that your specific contribution and goal in this piece is methodological. You now reach that point on page 3, line 110. It would be helpful to ensure your readers understand it as your focus in your abstract and introduction and thereafter that you make it central and its character clear throughout

Authors' reply:

We thank the reviewer for taking the time to comment a second time on our manuscript. We recognise the reviewer’s concern that we have not sufficiently emphasized the manuscript’s methodological contribution, which we acknowledge is a key focus. However, we believe that this focus is already sufficiently clear in the Abstract and Introduction. In support of this, we note that the Abstract clearly states that the manuscript’s focus is to investigate a potential source of inconsistency in the results of studies examining the relationship between drought and mental health, namely the use by those studies of a single spatiotemporal unit of analysis, which induces the MAUP and the MTUP (sentences 1-3). This focus is then elucidated in the Introduction, where we introduce the MTUP and the MAUP (paragraphs 5-6); suggest that examination of these problems might increase understanding of how variations and inconsistencies in previous studies’ results have occurred (paragraph 7); and describe how our manuscript will carry such an examination (paragraph 8). We note that, together, this material constitutes half of the entire Introduction. In Methods, the most extensive section, titled “Data preparation” (section 2.5), describes in detail the various data preparations which form the foundation of our investigation into the MAUP and the MTUP (see also Figs. 1-2). Finally, in Results, the section titled “Model results” (section 3.2) describes in detail the observed variation in estimates obtained using the different data preparations, and how we have combined these estimates to guide sensible conclusions and inference regarding the relationship between rainfall and MHED presentations (see also Figs. 6-8).

Having said this, we acknowledge that the methodological focus of the manuscript was not adequately reiterated in the Discussion. As such, we have made a number of revisions in that section in response to the reviewer’s comment. These revisions are too many to detail individually, but we refer the reviewer to the revised Discussion, in particular paragraphs 1-2, where the methodological focus of the paper is reiterated and relevant results are summarised.

Based on the above, we believe the methodological focus of the manuscript is sufficiently central and is made sufficiently clear to the reader throughout, and in particular in the Abstract and Introduction as the reviewer has suggested is necessary. If this response does not satisfy the reviewer, then we defer to the Editor.